# LLM Chemistry Estimation for Multi-LLM Recommendation

## Abstract

Multi-LLM collaboration promises accurate, robust, and context-aware solutions, yet existing approaches rely on implicit selection and output assessment without analyzing whether collaborating models truly complement or conflict. We introduce *LLM Chemistry* – a framework that measures when LLM combinations exhibit synergistic or antagonistic behaviors that shape collective performance beyond individual capabilities. We formalize the notion of chemistry among LLMs, propose algorithms that quantify it by analyzing interaction dependencies, and recommend optimal model ensembles accordingly. Our theoretical analysis shows that chemistry among collaborating LLMs is most evident under heterogeneous model profiles, with its outcome impact shaped by task type, group size, and complexity. Evaluation on classification, summarization, and program repair tasks provides initial evidence for these task-dependent effects, thereby reinforcing our theoretical results. This establishes LLM Chemistry as both a diagnostic factor in multi-LLM systems and a foundation for ensemble recommendation.

## 1 Introduction

Large Language Models (LLMs) are increasingly capable across tasks from code generation to open-domain question answering. Yet no single model excels universally (Chang et al., 2024). Different LLMs bring varied strengths—reasoning, generation, or domain expertise—spurring interest in multi-LLM collaboration: systems that coordinate multiple models to solve tasks collectively. By combining complementary abilities, such collaboration promises more accurate, robust, and context-aware solutions than any single model alone (Feng et al., 2024; Tran et al., 2025), though a key challenge lies in selecting ensembles that reliably exploit these strengths.

To address this challenge, a growing body of work has proposed numerous collaborative strategies positioned at different stages of the inference pipeline: (1) before inference, e.g., LLM routers (Rosenbaum et al., 2018); (2) during inference, e.g., ensemble decoding (Li et al., 2024a; Mavromatis et al., 2024); and (3) after inference, e.g., LLM Cascades (Chen et al., 2024b; Yue et al., 2024) and LLM compensatory cooperation (Zhao et al., 2024). These strategies aim to harness the strengths of multiple and frequently collaborating LLMs.

Existing methods focus primarily on selecting *strong individual models*—often large, high-performing closed-source LLMs such as ChatGPT, Claude, and Gemini— without considering how their interactions affect group performance (Hu et al., 2024a). Yet evidence shows these interactions can substantially influence collaborative outcomes (Liu et al., 2024) and even transmit unrelated behavioral traits via generated data (Cloud et al., 2025). Work on compound AI systems has examined LLM assignment across modules under resource constraints, but assumes modular independence and ignores potential synergy—or interference—when models collaborate on a single task.

To illustrate how such dynamics play out in practice, consider a *statement credibility classification* task: "*Small businesses create 70 percent of the jobs in America.* (Eric Cantor)." A single model may be precise but brittle (e.g., misclassifying unusual variations of this claim as *false*), while another may be broader but noisier (e.g., giving inconsistent answers such as *true, mostly true, half true*). Individually, neither suffices. Yet when models are combined and allowed to *interact through multiple rounds of response generation and critique*, they may converge on a more accurate answer—or entrench errors—showing that while multi-model solutions can outperform individual models, their collective performance depends as much on interaction as on individual accuracy (Du et al., 2023).

Motivated by this, we shift focus from allocation to interaction: rather than treating LLMs as isolated units, we study how they perform jointly on a single task, aiming to identify subsets that exhibit strong collaborative synergy—what we call *LLM Chemistry*. This raises a central question: ***For a given task, how can we identify which LLMs work best together, i.e., exhibit strong chemistry?***

To address this question, we propose a framework for quantifying LLM Chemistry among LLMs collaborating on shared tasks. We formalize this notion and present algorithms that identify inter-action dependencies among models and recommend those with strong chemistry. Informally, an LLM $a$ interacts with an LLM $b$ if the benefit of using $a$ on a task changes in the presence of $b$, and vice versa. We evaluate such chemistry purely through *performance evaluation*, i.e., how effectively LLMs achieve objectives together. Our experiments across three benchmarks provide initial evidence that interaction dynamics shape collective performance—sometimes amplifying, sometimes constraining. This suggests model selection alone may be insufficient, since outcomes also depend on interaction effects as well as individual strength. We model collaboration as a two-stage process of response generation and evaluation, a common setup in multi-LLM collaboration frameworks (Du et al., 2023; Madaan et al., 2023; Zhang et al., 2024). In all benchmarks, LLMs interact through this process rather than via independent subgoal assignment.

We summarize our technical contributions as follows:

- We introduce and formalize the notion of LLM Chemistry and present CHEME, an algorithm for computing the LLM chemistry within a group of LLMs.
- We introduce the concept of Model Interaction Graphs (MIGs) as a mechanism to encode the performance and cost tradeoffs of different LLM interactions in CHEME.
- We present RECOMMEND, an algorithm for selecting optimal LLM combinations that exhibit strong chemistry for collaborative tasks.
- We conduct empirical experiments across three diverse benchmarks demonstrating that interactions significantly influence collaborative performance and thus LLM Chemistry.

## 2 LLM CHEMISTRY

### 2.1 PRELIMINARIES

**Basic Concepts**. Let $Q$ be a query that needs answering. Suppose we have a group of candidate LLMs (denoted by $S$), each with recorded past performance (e.g., accuracy of LLMs and a quality score for their outputs). Our goals are: (1) to quantify how LLMs interact—their *chemistry*—when jointly answering $Q$; (2) to assess the impact of their interactions on performance; and (3) to identify the optimal *configuration* (i.e., subset of LLMs) for $Q$. Figure 2 illustrates this process.

Given a configuration $X \subseteq S$ chosen to answer $Q$, we introduce $cost_Q(X)$ as the total cost incurred by using $X$, and $used(X) \subseteq X$ as the subset of LLMs that successfully generated an answer for $Q$. Note that $X \setminus used(X)$ may be non-empty if some LLMs in $X$ did not contribute an output. The cost is defined as follows: assume $X$ produces $n > 0$ outputs for $Q$. Each output $o_i$ has a quality score $q_i \in [0, 10]$ (the consensus grade), each producing LLM has accuracy $a_i \in [0, 1]$ (its reliability score; see below for how $q_i$ and $a_i$ are computed), and a weight $w_i = 1/i$.[1] Since $q_i$ is in $[0, 10]$, we normalize it to $[0, 1]$ to match accuracy $a_i$: $q_i^{norm} = q_i/10$. Accordingly, we define the per-answer *penalty* (a measure of the joint error in quality and accuracy, which decreases as either improves) as $penalty_i = (1 - q_i^{norm})(1 - a_i)$. The total cost over all $n$ answers, weighted by $w_i$, is:

$$cost_Q(X) = \sum_{i=1}^{n} w_i \cdot penalty_i \tag{1}$$

The best-case scenario ($q_i^{norm} = 1, a_i = 1$): penalty is 0. Lower values in either quality or accuracy, increase the penalty (thus, the total cost). E.g., consider three answers with (rank, quality, accuracy): (1, 9.0, 0.90), (2, 8.0, 0.80), (3, 7.0, 0.70). Using weights $w_i = 1/i$, the total cost is $0.01 + 0.02 +$

---

[1]Weights are positive reals, inversely proportional to rank, so higher-ranked answers add more to the cost.

$0.0297 = 0.0597$. This shows low penalties for high-quality, high-accuracy answers, with penalties rising down the ranks; this cost analysis sets the stage for defining LLM benefit within an ensemble.

**Presence Benefit of an LLM in an Ensemble**. Given Equation 1 and disjoint sets $X, Y \subseteq S$, the *presence benefit* of an LLM is the change in performance cost when $Y$ is selected in addition to $X$,

**Definition 1** (Benefit). *Given two LLM sets $X$, $Y \subseteq S$ and a query Q, the benefit of $X$ with respect to $Y$ and Q is defined as $benefit_Q(X, Y) = cost_Q(Y) - cost_Q(X \cup Y)$.*

The value of $benefit_Q(X, Y)$ can be negative if adding $X$ to $Y$ raises the total cost, indicating degraded rather than improved performance. Understanding such degradations—and, more broadly, how LLMs affect each other's effectiveness—is key to analyzing model chemistry.

**Quality and Accuracy Scores**. We define the quality of an output $o_i$ (its consensus score $q_i$) as the aggregated estimate of its quality from multiple LLMs, and the accuracy of an LLM ($a_i$) as how closely its evaluations align with these consensus grades. We compute quality scores $q_i$ using a consensus-based approach inspired by the *Vancouver crowdsourcing algorithm* (de Alfaro & Shavlovsky, 2014). We apply the *Minimum Variance Linear Estimator* (MVLE) to iteratively infer consensus scores for each output and a consensus variance for each LLM. The consensus score becomes $q_i$, with higher values assigned to models whose outputs consistently align with low-variance consensus. In this setup, as in MVLE, the *inverse variance* serves as a proxy for *review accuracy*. *Generation accuracy* measures how well an LLM's outputs match ground truth reference answers, computed by direct comparison when available (values in $[0.0, 1.0]$), or set to $0.0$ when none exist.

To complement this, we compute a *continuous* accuracy $a_i$ for an LLM by combining generation and review accuracy. With ground truth, we weight them $75\%/25\%$; without it, $25\%/75\%$. For example, o3-mini on statement classification has generation accuracy $1.0$ and review accuracy $0.846$, yielding $a_i = 0.962$. We use MVLE over traditional voting to compute $q_i$ and $a_i$ for its statistical grounding, robustness to noise, and ability to refine reliability estimates.

### 2.2 The LLM Chemistry Problem

Beyond assessing individual contributions, we aim to understand how LLMs in a set $S$ interact when jointly answering a query $Q$. These interactions—whether cooperative or redundant—can significantly impact the overall effectiveness of the ensemble. Intuitively, two LLMs $a$ and $b$ interact when their individual benefits are dependent. This can occur in two ways. First, if $a$ and $b$ exhibit overlapping performance ($q_i, a_i$) profiles—i.e., they produce similar or redundant outputs on $Q$— then one can substitute for the other. Building on our notion of LLM chemistry, this constitutes a *negative chemistry*, as using both LLMs adds little benefit over using just one. Second, if $a$ and $b$ contribute complementary outputs—e.g., one extracts factual content while the other interprets it— their combination can yield a better answer than either alone. This constitutes a *positive chemistry*.

We formalize this intuition by defining the *LLM Chemistry* between two LLMs $a$ and $b$, denoted $chem_Q(a, b, S)$, as the change in the benefit of $a$ when $b$ is added to a set $X$:

**Definition 2** (LLM Chemistry). *Given LLMs a, b in S, and $X \subseteq S$ be a set of LLMs in S such that $X \cap \{a,b\} = \emptyset$, the chemistry between LLMs a and b w.r.t. Q is defined as:*

$$chem_Q(a, b, S) = \max_{X \subseteq S \setminus \{a,b\}} \frac{\Delta(a, b, X)}{cost_Q(X \cup \{a,b\})} \tag{2}$$

*where $\Delta(a, b, X)$ is the absolute difference between $benefit_Q(\{a\}, X)$ and $benefit_Q(\{a\}, X \cup \{b\})$.*

This definition reflects how strongly LLMs $a$ and $b$ complement each other when answering a query $Q$. Specifically, the LLM chemistry is highest when $a$ and $b$ provide highly complementary information, and lowest when their outputs are redundant, conflicting, or erroneous. To ensure that $chem(a,b)$ is independent from any specific choice of $X$, we define it explicitly as the maximum interaction effect across all possible subsets $X \subseteq S \setminus \{a,b\}$. This approach captures the worst-case interaction scenario, ensuring the measure reflects the strongest potential chemistry between two LLMs, independent of particular subset selections. For clarity, we omit $Q$ in $cost(X)$, $benefit(X, Y)$, and $chem_Q(a, b, S)$ when it is clear from context.

The LLM chemistry between any pair of LLMs is critical to understanding how the set $S$ influences answers to $Q$. This leads to a key problem addressed in this paper:

**Problem 1** (LLMs Chemistry Problem (**LLMCP**)). *Given a set of LLMs $S$, a subset $X \subseteq S$, and a threshold $\tau \geq 0.0$, find all pairs of distinct LLMs $a, b \in S$ such that $chem(a, b, S) > \tau$ w.r.t. $Q$.*

In this problem, we aim to identify pairs of LLMs that exhibit sufficiently strong chemistry for answering $Q$. The goal is not to compute exact chemistry values, but to decide whether chemistry is sufficiently high. Detecting strong chemistry requires finding only a single LLM configuration where threshold $\tau$ is exceeded, while ruling it out requires checking all relevant configurations. Thus, worst-case complexity remains as high as computing full LLM Chemistry.

## 3 CRITERIA FOR OPTIMAL LLM SELECTION

In our formulation of *LLMCP*, LLM interactions are evaluated by $cost(X)$ (Equation 1) to determine the optimal LLM selection for a query $Q$. This requires computing $chem(a, b)$ for every LLM pair $a, b$ across all subsets $X \subseteq S$, giving a lower bound of $\Omega(2^{|S|}, |S|^2)$.

In practice, the optimization of LLM selection should not depend on a particular set $X$ in an arbitrary way. For instance, adding more LLMs to the available set should not increase the cost of answering $Q$, since a larger set only broadens the space of possible responses. To avoid such anomalies, we make a natural assumption on $S$. Conceptually, we assume $S$ is made of diverse and independently strong LLMs whose strengths compensate for one another's weaknesses, rather than reinforce them. Informally, we say $S$ is **diverse** if its members exhibit heterogeneous performance characteristics— i.e., they differ meaningfully in their performance $(q_i, a_i)$ profiles. Conversely, $S$ is **homogeneous** if all models produce similar responses with comparable accuracy (i.e., aligned performance profiles).

This diversity assumption has profound implications for cost function behavior and thus on LLM Chemistry estimation. Under this assumption, $cost(X)$ should exhibit structure that reflects the reliability and diversity of the underlying models. Rather than treating all LLM combinations as equally unpredictable, we posit that cost varies predictably with model quality, accuracy, and set composition. This motivates three key properties of the cost function. These properties—monotonicity, linearity, and submodularity—not only align with practical expectations but also play a central role in making the selection of optimal LLM sets computationally feasible.

### 3.1 PROPERTIES OF THE COST FUNCTION

We formalize three properties that characterize Equation 1:

**Property 1** (Monotonicity). *The cost of a set $X$ for answering $Q$, $cost(X)$, is monotonically decreasing with the output quality scores $q_i$ and the accuracy values $a_i$ of the LLMs producing them.*

This property means the cost decreases as an LLM's accuracy or output quality increases, reflecting the intuition that accurate, high-quality outputs lower the total cost. (Proof in Appendix A.1.)

**Property 2** (Linearity of Cost). *The $cost(X)$ separates as a sum of terms, each depending only on the quality $q_i$ of an individual LLM output and the accuracy $a_i$ of its producing LLM.*

This property means the cost is additive: each LLM affects the total cost only through its own contribution, with no cross-terms or interactions between outputs. (Proof in Appendix A.1.)

**Property 3** (Submodularity). *For all sets $X \subseteq Y \subseteq S$ and any LLM $a \in S \backslash Y$, Equation 1 satisfies: $cost(X) - cost(X \cup \{a\}) \geq cost(Y) - cost(Y \cup \{a\})$.*

This property implies diminishing returns: for any given LLM $a$, its marginal benefit is larger when added to a smaller set $X$ than to a larger superset $Y$. (Proof in Appendix A.1.)

### 3.2 MODEL INTERACTION GRAPHS (MIGs)

To efficiently compute LLM chemistry, we introduce the *Model Interaction Graph* (MIG), a directed acyclic graph (DAG) over subsets of $S$. The MIG encoding draws inspiration from the *Index Benefit Graph* (IBG) (Frank et al., 1992). Figure 1 shows an MIG for $S = \{a, b, c\}$.

A central property of MIGs, inherited from IBGs, is that reasoning about subsets does not require every subset to be explicitly represented: larger nodes contain enough information to compute $cost(X)$ and $used(X)$ for overlapping subsets, allowing the MIG to compactly encode all necessary information and avoid exponential blowup. Each node in the MIG represents an LLM set $X \subseteq S$ and stores both $used(X)$ and $cost(X)$. Edges capture how costs change as LLMs are added or removed. Nodes and edges are constructed hierarchically, starting from the full set $S$; at each step, for each node $X$ and used LLM $a \in used(X)$, we create a new node $X' = X \setminus a$ with a directed edge from $X$ to $X'$, unless $X'$ already exists.

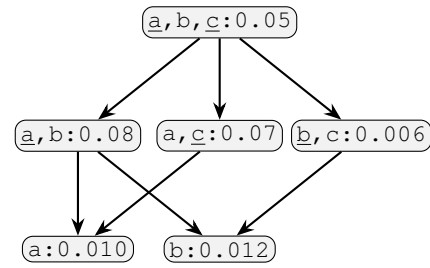

Figure 1: MIG for $S = \{a, b, c\}$. Underlined elements indicate $used(X)$ (LLMs with $a_i \geq 0.5$). Sample cost values are provided in each node.

## 4 COMPUTING LLM CHEMISTRY FOR MULTI-LLM RECOMMENDATION

### 4.1 COMPUTING LLM CHEMISTRY

In this section, we introduce CHEME (Algorithm 1). Given a query $Q$, CHEME builds a MIG and looks at its structure to compute $chem(a, b, S)$ for every pair of LLMs in $S$ w.r.t. $Q$. It does so by iterating over all subsets $X \subseteq S$ to compactly encode costs associated with various combinations of LLMs over relevant subsets of $S$. CHEME explicitly enumerates subsets to compute this maximum value, directly aligning with Definition 2.

For example, given the MIG in Figure 1, assume additional hypothetical costs $cost(\{c\}) = 0.15$ and $cost(\emptyset) = 0.20$, and the case where $X = \{c\}$. Using the MIG costs, we have $benefit(\{a\}, \{c\}) = 0.08$, $benefit(\{a\}, \{b, c\}) = 0.01$, yielding $chem(a, b, \{c\}) = \frac{|0.08 - 0.01|}{0.05} = 1.4$. As $\{c\}$ is the only relevant subset, the LLM chemistry for $(a, b)$ is also 1.4.

CHEME yields the following main result:

**Theorem 1.** *LLM Chemistry emerges in $S$ as a function of the MIG iff models exhibit heterogeneous performance $(q_i, a_i)$ profiles; for (near-)identically performing models, cost-based selection pressure vanishes, so no interaction effects can be detected (chemistry = 0).* (Proof in Appendix A.2).

---

**Algorithm 1:** Computes $chem(a, b, S)$ for all LLM pairs $a, b$ using memoized MIG lookups. $\mathcal{G}_Q$ encodes subset relationships. Solves **LLMCP**. Sets $\tau = 0$ to retain all chemistry information for later analysis.

**Function:** CHEME
**Input**   : A set of LLMs $S$.
**Output**  : The LLM chemistry for each distinct $a, b \in S$.
**Data**    : Hash table $t_Q : S \times S \to \mathbb{R}$. Memoized node-and-cost lookup.

1  Initialize $t_Q[a, b] \leftarrow 0$ for each distinct $a, b \subseteq S$
2  Construct the MIG $\mathcal{G}_Q$ for $Q$
3  **for** $k \leftarrow 0$ **to** $|S|$ **do**
4      **foreach** $X \subseteq S$ *with* $|X| = k$ **do**
5          Let $Y \supseteq X$ in $\mathcal{G}_Q$ and $cost(Y)$ be memoized
6          **if** $Y$ *is undefined* **then**
7              | **continue**
8          **foreach** *distinct* $a, b \in S - X$ **do**
9              Let $X_a \leftarrow X \cup \{a\}, X_b \leftarrow X \cup \{b\}, X_{ab} \leftarrow X \cup \{a, b\}$
10             Memoize $Y_a \supseteq X_a, Y_b \supseteq X_b, Y_{ab} \supseteq X_{ab}$ in $\mathcal{G}_Q$ and their costs
11             **if** *any of* $Y_a, Y_b, Y_{ab}$ *are undefined* **then**
12                 | **continue**
13             $d \leftarrow \frac{|benefit(\{a\}, Y) - benefit(\{a\}, Y_b)|}{cost(Y_{ab})}$
14             $t_Q[a, b] \leftarrow \max\{t_Q[a, b], d\}$
15             $t_Q[b, a] \leftarrow \max\{t_Q[b, a], d\}$

16 **return** $chem(a, b) = t_Q[a, b]$ *for each* $\{a, b\} \subseteq S$

---

This novel, counter-intuitive finding suggests that model diversity (i.e., model performance diversity) is essential for chemistry estimation: near-maximal performance makes chemistry-based multi-LLM recommendation unnecessary, while near-zero performance makes them infeasible.

A direct consequence of Theorem 1 is that chemistry scales with model performance diversity, as formalized in Corollary 1:

**Corollary 1.** *Chemistry in $S$ varies monotonically with model performance diversity: as diversity increases, chemistry either always increases or always decreases, depending on task type.* (Proof in Appendix A.2)

Theorem 1 and Corollary 1 establish the theoretical basis for LLM Chemistry, which we now operationalize in the next subsection to select effective ensembles for a given query, directly addressing the central question posed in the introduction.

### 4.2 MULTI-LLM RECOMMENDATION

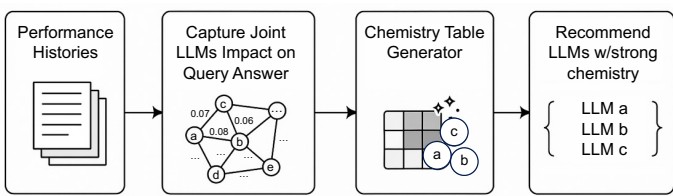

Figure 2: Illustration of LLM Chemistry estimation process for multi-LLM recommendation. A snapshot of the performance histories is provided in Appendix A.5.

We illustrate our LLM Chemistry estimation process for multi-LLM recommendation process in Figure 2. The figure outlines the main stages of a multi-LLM recommendation session, from an initial MIG construction for $Q$ from past performance histories to LLM Chemistry estimation, to the recommendation of an optimal LLM configuration. The central goal of this process is to recommend a set of LLMs that maximizes the benefit of collaboration while minimizing the cost of answering $Q$. To formalize this, we define $\mathcal{X} = \{X_1, X_2, \ldots, X_n\}$ as a collection of non-empty subsets of $S$, where each $X_i \subseteq S$ is a candidate LLM configuration. *Subsets may overlap, meaning the same LLM can appear in multiple configurations.*

To evaluate how well these subsets group related LLMs, we define a loss function $\mathcal{L}(\mathcal{X})$ that measures unrealized chemistry potential for $\mathcal{X}$. $\mathcal{L}(\mathcal{X})$ has two components: (1) inter-subset loss $\mathcal{L}_{inter}(\mathcal{X})$, the potential lost between LLMs placed into different subsets, and (2) intra-subset loss $\mathcal{L}_{intra}(\mathcal{X})$, the potential not realized within a subset compared to the theoretical maximum. Two parameters control these trade-offs: $\alpha \in [0, 1]$ balances the components, and $\beta > 0$ penalizes larger subsets to favor smaller, targeted recommendations (Equation 3).

$$\mathcal{L}(\mathcal{X}) = \alpha \cdot \mathcal{L}_{inter}(\mathcal{X}) + (1 - \alpha) \cdot \mathcal{L}_{intra}(\mathcal{X}) + \beta \cdot |\mathcal{X}| \tag{3}$$

$$\mathcal{L}_{inter}(\mathcal{X}) = \sum_{\substack{a,b \in S \\ a \neq b, \text{ different subsets}}} chem(a,b,S) - \sum_{\substack{X_i, X_j \in \mathcal{X} \\ i < j}} \sum_{a \in X_i} \sum_{\substack{b \in X_j \\ b \neq a}} chem(a,b,S) \tag{4}$$

$$\mathcal{L}_{intra}(\mathcal{X}) = \sum_{a < b \in S} chem(a,b,S) - \sum_{X_i \in \mathcal{X}} \sum_{a < b \in X_i} chem(a,b,S) \tag{5}$$

These loss terms capture unrealized chemistry from separation (inter-subset) and incomplete groupings (intra-subset[2]) of LLMs in $\mathcal{X}$. In this paper, we set $\alpha = 0.5$ to weight them equally and $\beta = 0.5$ to moderately penalize large subsets, typically recommending up to 10 LLMs.

Algorithm 2 recommends the optimal LLM subset from historically-derived configurations by leveraging pre-computation and the error function in Equation 3. Rather than exploring the exponential space of all possible subsets ($2^{|S|}$), it evaluates only $n$ historical subsets as starting points and performs local optimization through hill climbing with at most $iters$ iterations per starting point. This algorithm has a runtime complexity of $O(n \times iters \times |S|^3)$, which is polynomial in the input size and exponentially better than the brute-force approach of evaluating all subsets. This algorithm systematically explores the search space by starting from each historical subset in $\mathcal{X}$, and applying local search through single LLM additions, removals, or swaps to find a better configuration. It uses the LOSS function to evaluate each candidate solution and guide the optimization process. Specifically, LOSS computes intra-subset chemistry as $\sum_{a < b \in X} chem(a, b, S)$ and inter-subset chemistry as $\sum_{a \in X} \sum_{b \in S \setminus X} chem(a, b, S)$.

---

[2]In Equation 5, $a < b \in S$ denotes an arbitrary ordering, with each unordered pair $a, b$ counted once.

These become loss terms by subtracting from the theoretical maxima ($maxT$ and $maxI$, using all LLMs in $S$), then combining per Equation 3.

In homogeneous cases—i.e., (near-)identically performing models—LLM chemistry evaluates to zero by Theorem 1. Thus the RECOMMEND algorithm either selects a single LLM configuration above the threshold $\tau$ when all models are strong, or yields none if all are weak.

## 5 EXPERIMENTS

### 5.1 EXPERIMENT SETUP

We evaluate our approach on three key aspects: (1) ensemble effectiveness —whether chemistry-based selection outperforms baseline selection strategies in their respective tasks, (2) chemistry-complementarity correlation— how well chemistry scores predict *ensemble complementarity*[3], and (3) interaction effects across task complexity—how chemistry differs by task and difficulty.

**Datasets:** We evaluate on three benchmark datasets across diverse collaborative LLM scenarios: (1) *Liar benchmark* (Wang, 2017) for (English) statement credibility classification ($4,000$ statements with context and labels); (2) *MTS-Dialog benchmark* (Ben Abacha et al., 2023) for clinical notes summarization ($1,700$ doctor-patient conversations and notes); and (3) *Quixbugs benchmark* (Lin et al., 2017) for automated program repair (40 buggy code snippets with fixes).

**Algorithm 2:** Optimal LLM Subset Recommendation for $Q$.

---
**Function:** RECOMMEND
**Input** : LLMs $S$, query $Q$, factor $\alpha$, max iters.
**Output** : Optimal subset $X^*$ for $Q$.
1   $best \leftarrow \emptyset; minLoss \leftarrow \infty$
2   Get $\mathcal{X}_Q = \{X_1, \ldots, X_n\}$ from past runs
    // Precompute using CHEME
3   $maxT \leftarrow \sum_{a<b \in S} chem(a,b,S)$
4   $maxI \leftarrow \sum_{a,b \in S, a \neq b} chem(a,b,S)$
5   **foreach** $X_i \in \mathcal{X}_Q$ **do**
6     $cur \leftarrow X_i$;
     $loss \leftarrow \text{LOSS}(cur, maxT, maxI, \alpha)$
7     **if** $loss < minLoss$ **then**
8      $best \leftarrow cur; minLoss \leftarrow loss$
     // Local search via hill climbing
9     **for** $iter \leftarrow 1$ **to** $iters$ **do**
10      $bestN \leftarrow null; minN \leftarrow \infty$
11      **foreach** *neighbor* $X'$ *of cur* **do**
12       $lossN \leftarrow \text{LOSS}(X', maxT, maxI, \alpha)$
13       **if** $lossN < minN$ **then**
14        $bestN \leftarrow X'; minN \leftarrow lossN$
15      **if** $minN \geq loss$ **then**
16       **break**
17      $cur \leftarrow bestN; loss \leftarrow minN$
18      **if** $loss < minLoss$ **then**
19       $best \leftarrow cur; minLoss \leftarrow loss$
20   **return** $best$

---

**Task complexity:** Datasets span different domains, and we categorize their *task complexity* by considering both computational resources and reasoning effort. Following (Qi et al., 2025), we distinguish three levels of task complexities: low (minimal reasoning or compute), medium (extended reasoning or compute), and high (substantial reasoning or compute, often beyond model capability). Under this characterization, *Liar* is low, *MTS-Dialog* medium, and *Quixbugs* high.

**Models:** We evaluate on a diverse pool of LLMs spanning different model families and capabilities: GPT-4o, o1/3/4-mini, Claude Sonnet 3.5/3.7, Gemini 2.0 flash (closed-source models), and Llama 3.1/3.3 70B, Mixtral 8x22B, FireFunction v2, Qwen 2.5 32B (open-weights models). We selected these models to capture variation in architectures, skills, and resource demands across open- and closed-weight ecosystems, providing a representative basis for evaluation.

**Metrics:** We report (1) Ensemble Effectiveness and Improvement over Best Baseline - overall ensemble accuracy via soft voting over model accuracies (correct if mean accuracy $> 0.5$, majority rule), (2) Ensemble complementarity - extent of diverse strengths among ensemble members, and (3) Chemistry correlation coefficients measuring chemistry-complementarity relationships.

**Protocol:** For each benchmark, we evaluate all selection strategies across three group size configurations: $N = 3, 5, 10$ models. For each size configuration, we conduct 10 independent trials per selection strategy, with each trial executing across 10 records, resulting in $\approx 1,800$ total experimental task executions across all configurations (exact number varies due to likely execution failures). Selection strategies include: (1) Remote - closed-source models, (2) Local - open-weights models, (3) Random - randomly selected models, (4) Performance - top-k best individual performers, and

---

[3]We define ensemble complementarity as the extent to which LLM ensemble members contribute different strengths, providing better coverage of accuracy-quality trade-offs than any single model. We measure this using hypervolume (S-metric (Zitzler & Thiele, 2002)) and Rao's quadratic entropy (Rao, 1982), combined into a single index balancing performance coverage with member diversity.

(5) Chemistry - models recommended by our approach. Performance metrics are aggregated across tasks within each trial. Performance metrics are aggregated per trial, with ANOVA used for strategy comparisons and Pearson correlations for chemistry-complementarity, with significance at $p < 0.05$. These baselines follow LLM ensemble practice (Lu et al., 2024): random and top-k are standard, while remote vs. local reflects deployment constraints often discussed in applied studies.

## 5.2 RESULTS AND ANALYSIS

Table 1 evaluates whether chemistry-based selection improves ensemble effectiveness compared to baseline strategies. For statement credibility classification, chemistry yields measurable gains over baselines (+14.9% at $N = 5$, +2.6% at $N = 10$), with the largest gains in ensembles of size 5. For automated program repair, chemistry matches baseline effectiveness across group sizes (0.0%), reflecting ceiling effects where performance has saturated. For clinical note summarization, chemistry underperforms in small groups (−30.0% at $N = 3$) but converges to baseline in larger ensembles (+0.2% at $N = 5$, −0.1% at $N = 10$). These results indicate that chemistry's contribution to effectiveness is positive in some tasks and ensemble sizes, neutral in others, and negative when ensembles are small or saturated, underscoring its task- and size-dependent role in ensemble effectiveness.

Table 1: Comparison of Ensemble Effectiveness: Chemistry-Based vs. Baseline Selection (PERF.)

| Task | Group Size ($N$) | Chemistry | Best Baseline | Improvement ($\Delta\%$) |
|---|---|---|---|---|
| Statement Credibility Classification | 3 | 0.547 | 0.542 | +0.9% |
| | 5 | 0.680 | 0.592 | +14.9% |
| | 10 | 0.679 | 0.662 | +2.6% |
| Clinical Notes Summarization | 3 | 0.700 | 1.000 | −30.0% |
| | 5 | 1.000 | 0.998 | +0.2% |
| | 10 | 0.999 | 1.000 | −0.1% |
| Automated Program Repair | 3 | 1.000 | 1.000 | +0.0% |
| | 5 | 1.000 | 1.000 | +0.0% |
| | 10 | 1.000 | 1.000 | +0.0% |

**Note:** Ensemble effectiveness = Overall ensemble accuracy (via soft voting). Values are average ensemble effectiveness across all configurations for each group size, comparing chemistry-based selection with the best baseline. **Improvement ($\Delta\%$)** = Relative difference: (Chemistry − Baseline)/Baseline $\times$ 100. Chemistry-based = Algorithm 2's output. See Table 4 for model compositions of best ensembles.

Table 2 examines how well chemistry scores predict ensemble complementarity. For statement credibility classification, chemistry is positively associated with complementarity ($r = 0.319$, $p < 0.001$), suggesting that higher chemistry scores correspond to ensembles that cover the accuracy-quality trade-off more effectively.[4] By contrast, automated program repair ($r = −0.154$, $p < 0.01$) and clinical note summarization ($r = −0.226$, $p < 0.001$) show negative correlations, indicating that higher chemistry scores are linked to ensembles with reduced model performance diversity. These findings provide evidence that chemistry is a significant predictor of complementarity across all three tasks, but the direction of the relationship differs: in classification it signals beneficial complementarity, whereas in program repair and summarization it reflects reductions in performance diversity consistent with saturation effects where little additional complementarity can be realized.

Table 2: Chemistry-Ensemble Complementarity relationships by Task Type

| Task Type | Correlation ($r$) | Effect Size | p-value | Sig. | n |
|---|---|---|---|---|---|
| Statement Credibility Classification | 0.319 | Moderate | 0.000 | $***$ | 400 |
| Clinical Notes Summarization | −0.226 | Small-to-Moderate | 0.000 | $***$ | 394 |
| Automated Program Repair | −0.154 | Small-to-Moderate | 0.008 | $**$ | 291 |

**Note:** $r$ = Pearson's correlation coefficient. Results shown for strongest significant relationships per task type. Significance (Sig.) levels: $***$ $p < 0.001$, $**$ $p < 0.01$, $*$ $p < 0.05$. *Key finding:* LLM-Chemistry-ensemble complementarity relationship reverses by task type ($r = −0.154$ vs $r = 0.319$).

We treat task complexity as an orthogonal lens to the task-type categories to examine whether chemistry effects scale with difficulty (Table 3). For statement credibility classification (low complexity), chemistry scores correlate positively with complementarity as group size increases ($r = 0.118$, $p < 0.05$ for $N = 5$; $r = 0.319$, $p < 0.001$ for $N = 10$), while effectiveness shows no significant association. For clinical note summarization (medium complexity), chemistry shows no

---

[4]Due to space limitations, visualizations of these trade-offs are provided in Appendix A.4.

association in smaller groups but correlates positively with complementarity at $N = 10$ ($r = 0.148$, $p < 0.01$). For automated program repair (high complexity), chemistry scores correlate negatively with complementarity in medium groups ($r = -0.154$, $p < 0.01$) but positively with effectiveness across all group sizes ($N = 3$: $r = 0.189$, $p < 0.001$; $N = 5$: $r = 0.144$, $p < 0.05$; $N = 10$: $r = 0.229$, $p < 0.001$). These results indicate that task complexity moderates chemistry's role: in low-complexity tasks, chemistry enhances complementarity without affecting effectiveness; in medium-complexity tasks, it begins to matter only for larger ensembles; and in high-complexity tasks, chemistry reduces complementarity but consistently improves effectiveness.

Table 3: Task Complexity and Chemistry Effects on Ensemble Performance

| Task (Complexity) | Group Size ($N$) | Complementarity ($r$) | Sig. | Effectiveness ($r$) | Sig. | n |
|---|---|---|---|---|---|---|
| Statement Credibility Classification (Low) | 3 | 0.089 | ns | 0.014 | ns | 368 |
| | 5 | 0.118 | $*$ | $-0.068$ | ns | 358 |
| | 10 | 0.319 | $* * *$ | $-0.024$ | ns | 400 |
| Clinical Notes Summarization (Medium) | 3 | 0.000 | ns | 0.000 | ns | 383 |
| | 5 | 0.000 | ns | 0.000 | ns | 357 |
| | 10 | 0.148 | $**$ | 0.027 | ns | 409 |
| Automated Program Repair (High) | 3 | $-0.043$ | ns | 0.189 | $* * *$ | 451 |
| | 5 | $-0.154$ | $**$ | 0.144 | $*$ | 291 |
| | 10 | $-0.086$ | ns | 0.229 | $* * *$ | 335 |

**Note:** Values are Pearson's $r$ correlations between chemistry scores and ensemble complementarity (Cols. 3-4) or effectiveness (Cols. 5-6). $n$ = number of ensembles. Significance (Sig.): $* * * \, p < 0.001$, $** \, p < 0.01$, $* \, p < 0.05$, ns = not significant. _Key finding:_ Chemistry-ensemble correlations vary with task complexity: complementarity dominates at low/medium complexity, effectiveness at high complexity.

Together, the results from Tables 1-3 show that LLM chemistry's contribution to ensembles is not uniform: (1) it improves effectiveness in some tasks and sizes, (2) predicts complementarity in a task-dependent manner, and (3) is further moderated by task complexity and group size.

## 6 RELATED WORK

As multi-AI systems become increasingly prevalent, researchers have explored **ensemble methods** for aggregating model outputs (Lu et al., 2024; Daheim et al., 2024; Zhang et al., 2024; Chen et al., 2025b) and solving various problems through model collaboration (Miao et al., 2024; Zhou et al., 2024; Feng et al., 2024; Zhao et al., 2024; Tran et al., 2025), **multi-agent communication topologies** for improving operation efficiency and effectiveness (Li et al., 2023; Hu et al., 2024b; Li et al., 2024b; Chen et al., 2024c), **automated selection techniques** for choosing optimal models (Rosenbaum et al., 2018; Chen et al., 2024b; Yue et al., 2024; Dekoninck et al., 2025; Ong et al., 2025; Patidar et al., 2025), and **compound AI frameworks** for orchestrating models across pipeline stages (Chen et al., 2024a; Santhanam et al., 2024; Chen et al., 2025a; Chaudhry et al., 2025; Wang et al., 2025). However, these approaches typically assume model independence or modular separation, overlooking the interaction dynamics that emerge when models collaborate on shared tasks. Unlike these methods, our work explicitly models the interaction effects between LLMs when they collaborate on shared tasks, introducing the novel concept of LLM chemistry to capture both synergistic and antagonistic relationships that influence collaborative performance.

## 7 CONCLUSIONS AND FUTURE WORK

We introduced the _LLM Chemistry_ estimation framework for multi-LLM recommendations that adapts to task demands. We found that chemistry can improve ensemble effectiveness in some tasks and sizes, diminish under ceiling effects where performance has saturated, and vary systematically with task complexity. These structured, non-uniform outcomes are consistent with our theoretical framework, showing that chemistry emerges under heterogeneous performance profiles and manifests differently by task type, ensemble size, and complexity. LLM chemistry estimation can be a _meta-learning signal_ for effective ensemble formation, indicating when chemistry is likely to help and when it is not. Looking forward, chemistry-aware modeling may improve next-generation architectures, from Mixture-of-Experts (MoE) and multi-agent systems to human-machine collaboration. Alongside this, linking individual LLM skills to chemistry could explain ensemble performance beyond the current "sum of parts" view.

ETHICS STATEMENT

Our work introduces the notion of LLM Chemistry, a framework for assessing LLM teaming capabilities and recommending optimal ensembles for reliable collaborative task completion. We demonstrate its effectiveness across state-of-the-art models and well-established benchmarks covering tasks of increasing difficulty.

By moving beyond the assumption that only large, high-performance closed-source LLMs guarantee strong performance, our approach enables ensembles that combine both open- and closed-weight models, thereby broadening accessibility and promoting fairness in multi-LLM systems.

We also recognize that LLM ensembles increase computational and energy costs, which must be weighed against their potential gains in robustness. By guiding efficient ensemble formation, our framework can help mitigate these costs and support sustainability. Finally, while our methods aim to improve the reliability of outputs, they could be misused to create more persuasive but deceptive systems; we discourage such applications and stress the importance of transparency in deployment.

REPRODUCIBILITY STATEMENT

The proposed LLM Chemistry pipeline is designed to be easily reproducible. We provide algorithms for computing chemistry (Algorithm 1) and for identifying and recommending optimal subsets of LLMs (Algorithm 2). Proofs and theoretical analyses are included in the Appendix due to page limitations.

To facilitate replication, we will release the full source code of the proposed LLM Chemistry. The repository will include a detailed `README.md` with setup instructions and scripts to reproduce all experiments. Running the code will require access to the datasets and LLMs described in Section 5.1. The open-weight models are available via the Ollama[5] platform, while closed-weight models require API access. Links to all the benchmark datasets used in this paper are provided in the References.

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

# A APPENDIX

## A.1 PROOFS OF COST FUNCTION PROPERTIES

We provide detailed proofs for the three properties of the cost function presented in Section 3.1.

**Proof of Property 1 (Monotonicity)**

*Proof.* Assume weights $w_i \geq 0$, each term contributing to the cost is of the form $w_i \cdot (1 - q_i^{norm})(1 - a_i)$. This expression decreases as either the output quality $q_i^{norm}$ increases or the LLM accuracy $a_i$ increases, with the other held fixed. Since the total cost is a weighted sum of such terms it is monotonically decreasing in both $q_i^{norm}$ and $a_i$. □

**Proof of Property 2 (Linearity)**

*Proof.* By Equation 1, the total cost is defined as: $cost(X) = \sum i = 1^n w_i \cdot (1 - q_i^{norm}) \cdot (1 - a_i)$. Each term depends solely on the quality score $q_i$ of single output $o_i$ and the accuracy $a_i$ of the LLM that produced it. There are no terms involving combinations of multiple outputs or LLMs. Therefore, the total cost is a linear sum of independent contributions, one per output-LLM pair, as claimed. $\square$

**Proof of Property 3 (Submodularity)**

*Proof.* The cost $cost(X)$ is a weighted sum of penalties from the outputs produced by the LLMs in $X$. Adding a new LLM $a$ to $X$ can only improve or maintain the best available outputs, thus $cost(X \cup \{a\}) \leq cost(X)$. Since $X \subseteq Y$, the set $Y$ already has more LLMs and thus better or more redundant outputs. As a result, the marginal benefit of adding $a$ to $Y$ is smaller than the benefit of adding $a$ to $X$. Formally,

$$cost(X) - cost(X \cup \{a\}) \geq cost(Y) - cost(Y \cup \{a\}).$$

Thus, the cost function is submodular. $\square$

A.2 THEORETICAL ANALYSIS OF CHEMISTRY EMERGENCE

Having established the algorithmic framework for computing LLM chemistry, we now ask a fundamental question: *when does chemistry emerge in model ensembles?* Answering this is critical for understanding the conditions under which chemistry-based optimization is feasible and beneficial.

Our analysis reveals a surprising and counter-intuitive result: chemistry emergence is intrinsically linked to model performance diversity, and perfect models paradoxically eliminate the very interactions that chemistry-based methods seek to exploit. This finding has profound implications for ensemble design and optimization strategy selection.

We establish our main theoretical contributions through two key results. First, we prove that chemistry emergence is equivalent to ensemble performance heterogeneity, while homogeneous ensembles yield zero chemistry regardless of individual model strength. Second, we show that chemistry scales monotonically with performance diversity, though the optimal level varies by task requirements. We now formalize our first result by proving Theorem 1.

*Proof.* If all models share the same performance $(q_i^*, a_i^*)$ profile, then each has identical penalty $p^* = (1 - q_i^{norm*})(1 - a_i^*)$. By the linearity property, $cost(X) = p^* \sum_{i=1}^{k} w_i$ depends only on set size, so $benefit(\{a\}, X)$ depends only on $|X|$. Adding $b$ changes only the set size, not the marginal effect of $a$. Thus

$$\Delta(a, b, X) = 0 \quad \Rightarrow \quad chem(a, b, S) = 0.$$

If models' performance differ, penalties vary, so adding $a$ or $b$ changes costs in model-specific ways. Then $benefit(\{a\}, X) \neq benefit(\{a\}, X \cup \{b\})$ for some $X$, giving $\Delta(a, b, X) > 0$. $\square$

We next prove Corollary 1, linking chemistry growth to model performance diversity.

*Proof.* By monotonicity, diverse performance $(q_i, a_i)$ profiles create diverse penalties, and by submodularity, this diversity has diminishing returns. Greater performance diversity increases variance in penalties, $\text{Var}[penalty_i]$. This amplifies selective pressure, producing more variation in benefits across subsets. Since LLM chemistry depends on

$$\max_X \frac{\Delta(a, b, X)}{cost(X \cup \{a, b\})},$$

larger benefit differences yield larger $\Delta$, hence greater chemistry. The precise "beneficial" diversity depends on task demands: high for complementary reasoning, lower for factual consistency. $\square$

## A.3 REPRESENTATIVE ENSEMBLES: CHEMISTRY VS PERFORMANCE BASELINE

Table 4 reports representative ensembles for each task and group size, showing the model compositions and effectiveness of the best observed LLM configurations (yielded by Algorithm 2) compared with the one yielded by the Performance baseline. These examples complement the average results in Table 1 by highlighting which specific model combinations achieved peak effectiveness.

Table 4: Best observed ensembles (Chemistry vs Best (Performance) baseline) for Statement Classification, Automated Program Repair, and Clinical Note Summarization. Each ensemble corresponds to the configuration achieving the highest single-run effectiveness.

| Task | Group Size | Chemistry (models) | Effectiveness | Best Baseline (PERF, models) | Effectiveness |
|---|---|---|---|---|---|
| Statement Credibility Classification | 3 | o4-mini
mixtral:8x22b
o1-mini | 0.600 | o4-mini
claude-3-5-sonnet-latest
o1-mini | 0.600 |
| | 5 | gpt-4o
gemini-2.0-flash
claude-3-5-sonnet-latest
claude-3-7-sonnet-20250219
firefunction-v2 | 0.700 | claude-3-5-sonnet-latest
claude-3-7-sonnet-20250219
o4-mini
qwen2.5:32b | 0.700 |
| | 10 | firefunction-v2
claude-3-7-sonnet-20250219
o1-mini
gpt-4o
gemini-2.0-flash
claude-3-5-sonnet-latest
qwen2.5:32b
o4-mini
o3-mini | 0.800 | o4-mini
gemini-2.0-flash
o3-mini
gpt-4o
claude-3-7-sonnet-20250219
o1-mini
claude-3-5-sonnet-latest
firefunction-v2
qwen2.5:32b | 0.800 |
| Clinical Notes Summarization | 3 | firefunction-v2
gemini-2.0-flash
claude-3-7-sonnet-20250219 | 0.700 | mixtral:8x22b
gpt-4o
qwen2.5:32b | 1.000 |
| | 5 | firefunction-v2
gemini-2.0-flash
qwen2.5:32b
gpt-4o
mixtral:8x22b | 1.000 | gpt-4o
o3-mini
llama3.1:70b
mixtral:8x22b
qwen2.5:32b | 1.000 |
| | 10 | mixtral:8x22b
firefunction-v2
gpt-4o
o4-mini
o3-mini
gemini-2.0-flash
claude-3-5-sonnet-latest
llama3.3:latest
o1-mini
qwen2.5:32b | 1.000 | gemini-2.0-flash
o3-mini
mixtral:8x22b
o1-mini
llama3.1:70b
firefunction-v2
gpt-4o
o4-mini
llama3.3:latest
qwen2.5:32b | 1.000 |
| Automated Program Repair | 3 | firefunction-v2
claude-3-7-sonnet-20250219
claude-3-5-sonnet-latest | 1.000 | gpt-4o
claude-3-5-sonnet-latest
o4-mini | 1.000 |
| | 5 | o3-mini
claude-3-7-sonnet-20250219
gpt-4o
mixtral:8x22b
llama3.1:70b | 1.000 | o1-mini
firefunction-v2
gpt-4o
claude-3-5-sonnet-latest
o4-mini | 1.000 |
| | 10 | o1-mini
firefunction-v2
o3-mini
claude-3-7-sonnet-20250219
qwen2.5:32b
gpt-4o
claude-3-5-sonnet-latest
mixtral:8x22b
llama3.1:70b
o4-mini | 1.000 | o1-mini
firefunction-v2
o3-mini
claude-3-7-sonnet-20250219
qwen2.5:32b
gpt-4o
claude-3-5-sonnet-latest
mixtral:8x22b
o4-mini
llama3.3:latest | 1.000 |

**Note:** For Clinical Note Summarization (group size 3), Chemistry dropped (0.700 vs. 1.000) as the `claude-3-7-sonnet-20250219` model underperformed. This is related to temporary service issues or budget limits at the time of evaluation.

## A.4 INTERPRETIVE ANALYSIS OF CHEMISTRY EMERGENCE

Having established Theorem 1 and Corollary 1, we now provide a visual interpretation through chemistry maps, which illustrate how the predicted conditions for LLM chemistry emergence manifest across the accuracy-quality space. These maps display the marginal complementarity ($\Delta$CI) of adding a new model to an ensemble, serving as a visual proxy for the theorem's guarantees. In particular, our theorem and corollary link chemistry to model performance heterogeneity: homogeneity implies $\Delta = 0$ and thus $chem = 0$, while heterogeneity implies $\Delta > 0$ and thus $chem > 0$. The maps make this relationship visible—bright regions mark conditions under which chemistry would emerge ($\Delta > 0$, heterogeneity), whereas dark regions reflect redundancy ($\Delta = 0$, homogeneity). Intuitively, ensembles with many bright regions have substantial room for chemistry to emerge and improve performance, ensembles with only tiny bright slivers allow for marginal gains at best, and fully dark maps indicate *saturation*, where new members cannot increase CI and performance gains from chemistry are no longer possible. In rare cases, the maps appear nearly uniform with negligible variation in $\Delta$CI; these are interpreted as *saturated ensembles* and not as evidence of broad chemistry. We present chemistry maps for three tasks of varying complexity:

## A.5 BENCHMARK SAMPLE RUNS

Each benchmark run produces a `CSV` file that records the outputs and metrics of LLM executions. The file contains one row for each (*trial*, *model*, *task*) combination, together with the performance and metadata fields listed in Table 5. A collection of such files constitutes our *performance histories* (See Figure 2's first step).

Table 5: Columns present in the CSV file produced by benchmark runs.

| Column | Data Type | Description |
|---|---|---|
| trial | string | Name or identifier of the benchmark run. |
| model | string | Name of the evaluated LLM. |
| task | string | Input task text (prompt or query $Q$). |
| latency | float | Time in seconds to generate the LLM output. |
| temperature | float | Sampling temperature used for generation. |
| id | string | Unique identifier for the generated output. |
| result | string | Raw text output produced by the LLM. |
| quality | float | Consensus-based quality score of the generated output. |
| gen_accuracy | float | Generation accuracy of the initial generation step. |
| variance | float | Variance of quality scores (dispersion/confidence measure). |
| review_accuracy | float | Review accuracy estimated via MVLE (inverse of variance). |
| accuracy | float | Final accuracy score, combining generation and review. |
| elapsed | string | Human-readable elapsed generation time. |
| created | string | Human-readable timestamp of output creation. |

**Task:** *Classify a short statement into a category of fakeness*

---

**model:** gpt-4o

**result:** "The Chicago Bears have had more starting quarterbacks in the last 10 years than the total number of tenured (UW) faculty fired during the last two decades." is labeled as "true".
**latency:** 2.0
**quality:** $1 \times 10^{-10}$
**gen_accuracy:** 0.0
**review_accuracy:** 0.9999999998482016
**accuracy**: 0.2499999999620504

---

**model:** o3-mini

**result:** "'I'm the only person on this stage who has worked actively just last year passing, along with Russ Feingold, some of the toughest ethics reform since Watergate.' is labeled as 'half-true'."
**latency:** 31.0
**quality:** 7.1397913333
**gen_accuracy:** 1.0
**review_accuracy:** 0.8969102035156231
**accuracy**: 0.9742275508789058

---

**model:** llama3.1:70b

**result:** "'However, it took $19.5 million in Oregon Lottery funds for the Port of Newport to eventually land the new NOAA Marine Operations Center-Pacific.' is labeled as 'true'."
**latency:** 82.0
**quality:** 1.9353739253
**gen_accuracy:** 0.0
**review_accuracy:** 0.224749679737892
**accuracy**: 0.056187419934473

---

**model:** gemini-2.0-flash

**result:** "'The economic turnaround started at the end of my term.' is labeled as 'half-true'."
**latency:** 2.0
**quality:** 9.0630267113
**gen_accuracy:** 1.0
**review_accuracy:** 0.8617875069130844
**accuracy**: 0.9654468767282711

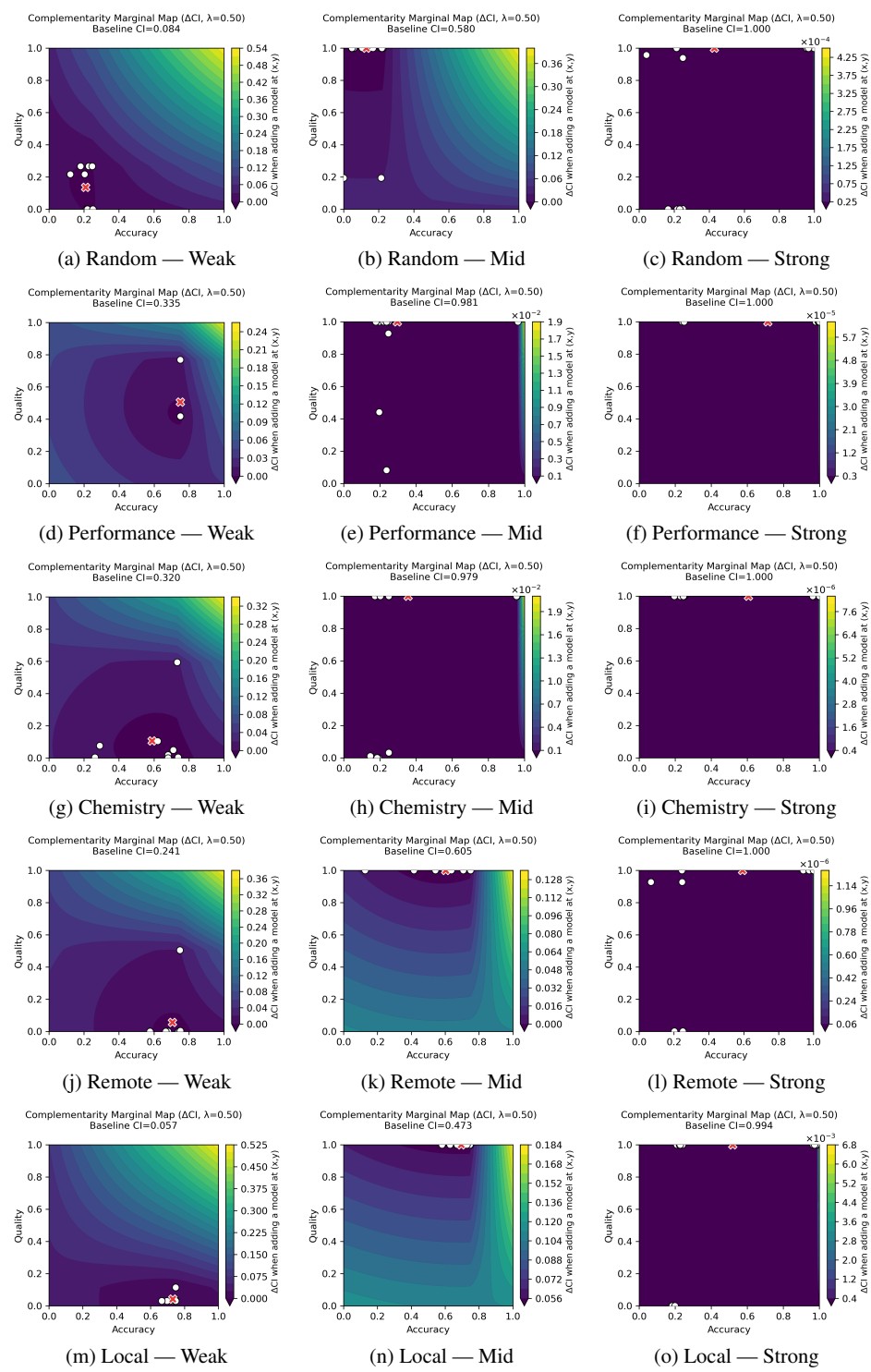

Figure 3: LLM chemistry maps (marginal complementarity, $\Delta$CI, trade-off parameter $\lambda = 0.5$) for **Statement Credibility Classification** (*low complexity*, $N = 10$). Rows correspond to strategies (Random, Performance, Remote, Local); the Chemistry row is included for comparison. Columns show **Weak**, **Mid**, and **Strong** ensembles. Weak ensembles display extensive bright regions, indicating substantial chemistry potential and performance gains. Mid ensembles are mixed, with some retaining bright regions and others already saturated. Strong ensembles are almost entirely dark, reflecting saturation where added models are redundant. A few weak panels appear nearly uniform in $\Delta$CI; these reflect negligible variation and are treated as saturated rather than as broad chemistry.

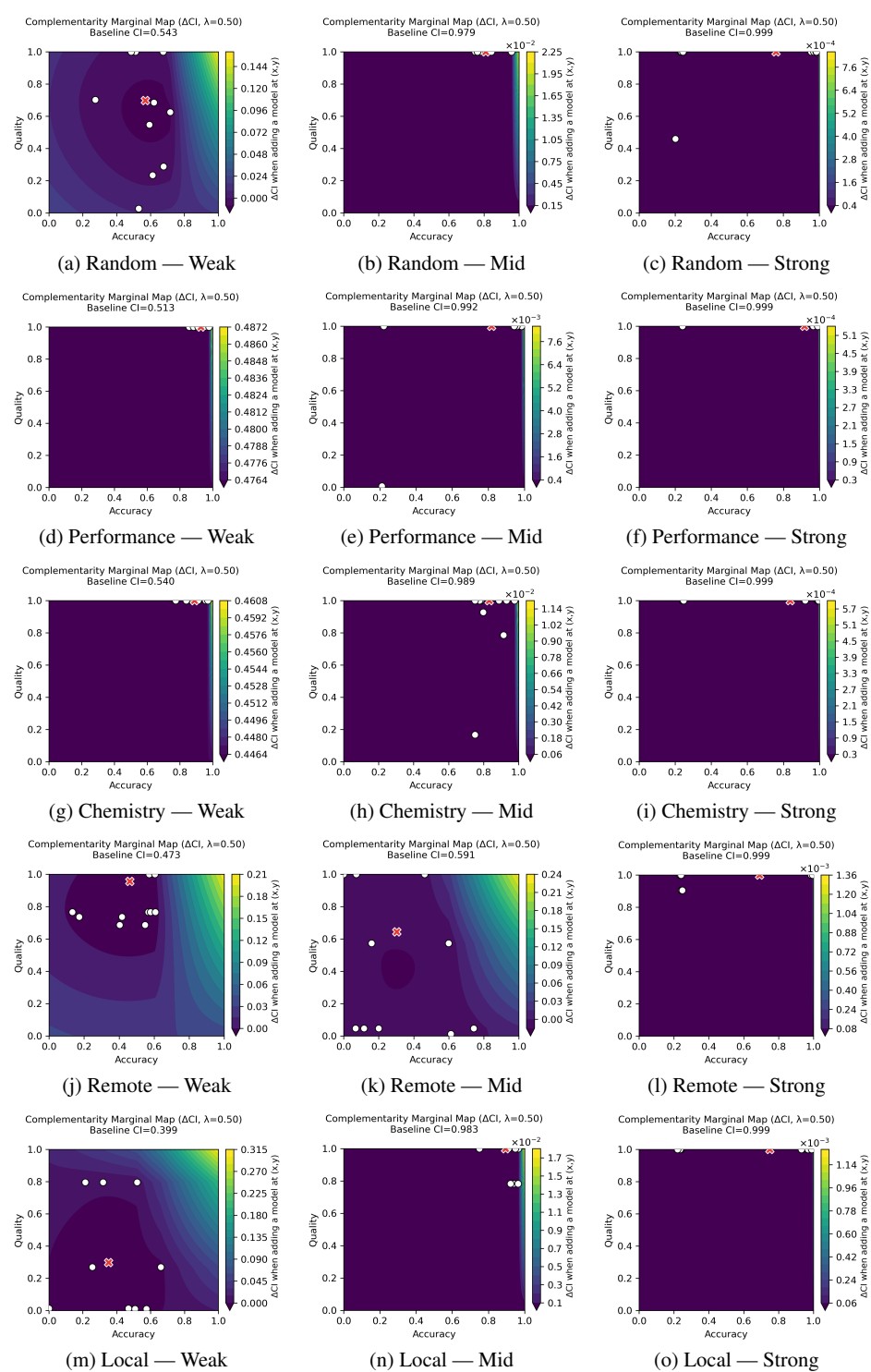

Figure 4: LLM chemistry maps (marginal complementarity, $\Delta$CI, trade-off parameter $\lambda = 0.5$) for **Clinical Notes Summarization** (*medium complexity*, $N = 10$). Rows correspond to strategies (Random, Performance, Remote, Local); the Chemistry row is included for comparison. Columns show **Weak**, **Mid**, and **Strong** ensembles. Weak ensembles are mixed: three maps display bright regions (chemistry emergence possible), while two are mostly dark with marginal potential. Mid ensembles are largely dark, with one strategy retaining notable bright regions but most showing only small gains. Strong ensembles are uniformly dark with tiny slivers, indicating near-complete saturation where added models provide little benefit.

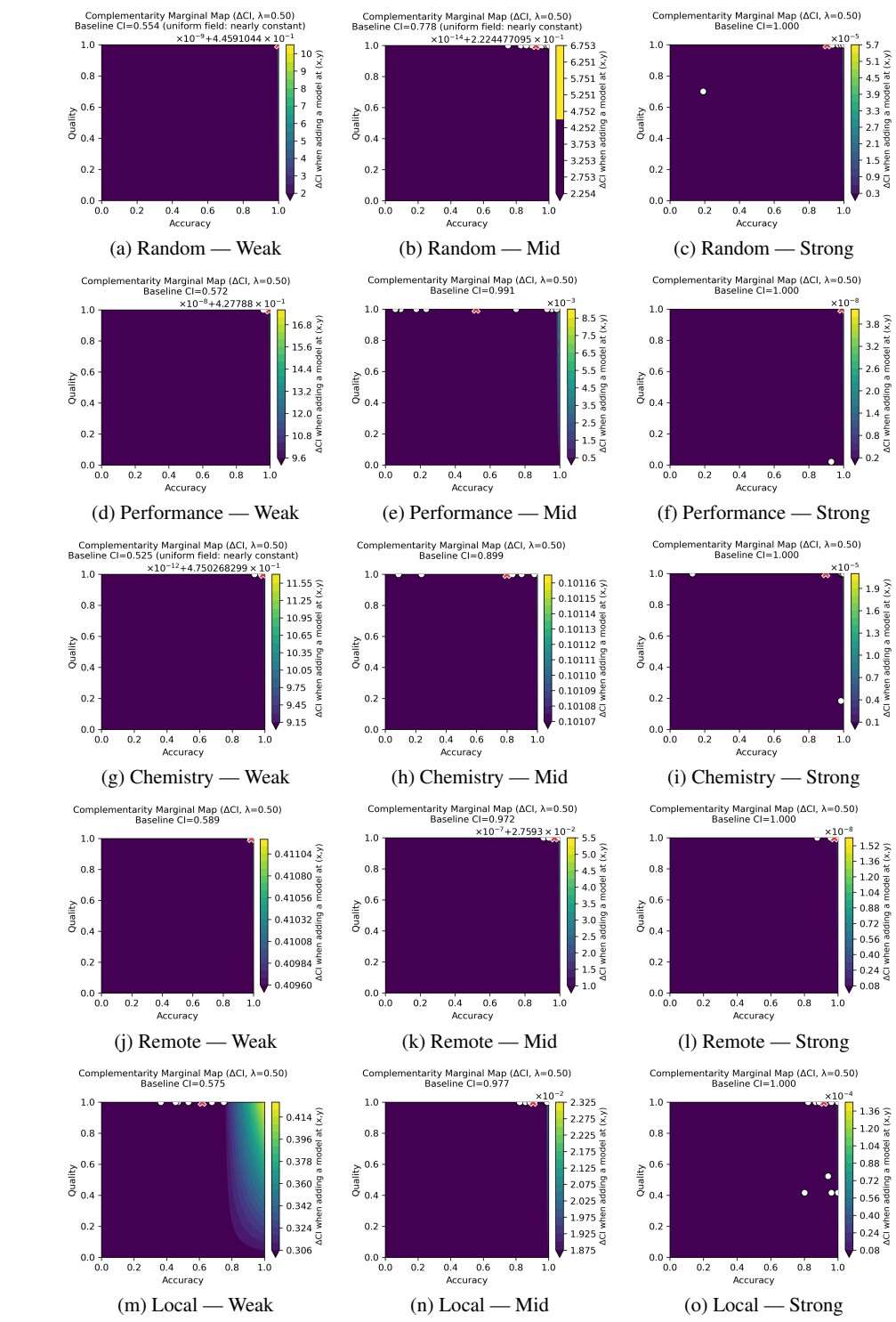

Figure 5: LLM chemistry maps (marginal complementarity, $\Delta$CI, trade-off parameter $\lambda = 0.5$) for **Automated Program Repair** (*high complexity*, $N = 10$). Rows correspond to strategies (Random, Performance, Remote, Local); the Chemistry row is included for comparison. Columns show **Weak**, **Mid**, and **Strong** ensembles. Bright regions are nearly absent across all ensembles, indicating saturation ($\Delta$CI $\approx 0$) where added models are redundant, chemistry emergence is negligible, and performance plateaus. Occasional uniform panels with vanishing $\Delta$CI variation are likewise interpreted as saturated.

