# OpenReview forum: "LLM Chemistry Estimation for Multi-LLM Recommendation"
_ICLR.cc/2026/Conference — Submitted to ICLR 2026_

### Official Review · Reviewer_UtP3 · 2025-10-26

**Soundness:** 3
**Presentation:** 3
**Contribution:** 3
**Rating:** 6
**Confidence:** 4

**Summary:**

This work proposes to quantify the "chemistry" between LLMs when used in collaboration, and motivate an approach for selecting models for a given task.

**Strengths:**

+ model collaboration is an important research direction
+ the problem formulation is interesting

**Weaknesses:**

- How do we estimate the max in equation 2? If the model set S is very large (for example, there are now >2m LMs on huggingface), is there an estimation/sampling step that we could take here?

- On line 187 seems to be that the formulation focuses on "pairs of LMs". I'm sure there are ways to extend this formulation beyond pairwise estimation, if the authors could share more.

- So the "model diversity" here is only considered as "diverse performance profiles". Would other "diversity" measurements, such as training data, architecture, or item response theory [1] be possible/better?

- I want to like this paper and it is very interesting, however, I have to say that in many places, very strong/unrealistic assumptions are made to support the theoretical framing. For example, the "pair of LMs" thing above. For property 2 on page 4, the output/cost of one LM could certainly affect another's (one model convinced another model of a wrong answer, one larger model generating feedback for a smaller model's answer), and assuming "no cross-terms" is perhaps a stretch. For property 3 on page 4, what if the superset Y is X + a low-quality/malicious LM? In this way, the performance under Y could be worse than X, and adding a new LM to it wouldn't guarantee a larger benefit. Perhaps I misunderstood something, or this is not a thing at all. I understand all problem formulations need to make some assumptions to make things smooth.

- One major concern is that the evaluation datasets (line 348) are unconventional for evaluating modern (2024-) language models. Things like QA, instruction following, reasoning, math, knowledge & factuality, safety, etc., come to mind, but they were not very covered here. The datasets are also sort of old/not the most popular ones. Perhaps the authors are not from a traditional NLP/LLM background, and perhaps this is fine.

- All models employed in the experiments (lines 360-364) are general-purpose industry models. Would they really be "performance diverse", which is critical in the success by Theorem 1? Something like [2-3] seems to suggest otherwise.

[1] Chen, Jianhao, et al. "Learning Compact Representations of LLM Abilities via Item Response Theory."

[2] Zhang, Lily Hong, et al. "Cultivating pluralism in algorithmic monoculture: The community alignment dataset."

[3] Jiang, Liwei, et al. "Artificial Hivemind: The Open-Ended Homogeneity of Language Models (and Beyond)."

**Questions:**

please see above

---

### Official Review · Reviewer_qnij · 2025-10-27

**Soundness:** 1
**Presentation:** 3
**Contribution:** 2
**Rating:** 2
**Confidence:** 4

**Summary:**

This paper addresses the challenge of forming effective multi-LLM ensembles by shifting the focus from individual model selection to analyzing collaborative interactions. Its central research question is: For a given task, how can we identify which LLMs work best together? Then, they introduce a framework designed to quantify the synergistic or antagonistic interactions between models when they collaborate on a shared task. They propose the ChemE and RECOMMEND algorithms to compute chemistry and recommend optimal ensembles and demonstrate its impact on performance on 3 distinct tasks.

**Strengths:**

1)  Research goal and objectives are clear since the intro with a clear research question that is well motivated and sound.

2)  Treating the problem as the combination of models modeling their interactions is crucial, and I agree with the author that it should be the way to go instead of evaluating each individually

3)  Provides a solid theoretical foundation by formally defining chemistry, a cost function, and its key properties (monotonicity, linearity, submodularity), supported by mathematical proofs.

4)  Complements the theory with practical algorithms (ChemE, Recommend) that obtain great results compared to other selection heuristics (e.g., top k)

**Weaknesses:**

1) As an ensemble selection model, which is conducted for each query (i.e., the best model for answering a particular instance). The authors must evaluate their performance against the ideal selection: The oracle (which is an abstract model that looks at the test set).

2) The paper does not mention the specific prompts used or detail any prompt engineering strategy for the individual LLMs in their ensembles. How did they set each model (open and closed-source) effectively for each task? This point can have a significant impact [4]. The problem is even bigger as no code & data were published for review. This makes it difficult to assess whether the observed chemistry is a true property of the models or an artifact of suboptimal prompting.

3) No source code availability, which is a major negative point, as we can’t assess whether the results are correct. This is a major flaw in reproducibility.

4) The proposed "chemistry" metric is presented in isolation without contextualizing it within the rich body of ensemble selection literature. This field has long-established metrics for quantifying model complementarity, such as prediction diversity (e.g., Q-statistics, double-fault), and frameworks for multi-objective optimization that balance accuracy with cost or diversity. The paper fails to demonstrate how "chemistry" relates to or improves upon these existing measures, or why a new metric was necessary. This lack of comparative analysis, especially against static (pre-computed) and dynamic (instance-specific) ensemble selection baselines, makes it difficult to assess the true novelty and contribution of the concept. At the end I am not entirely sure whether this should be really considered a new concept or just a rebranding of existing ones from a different literature/ research community.

5) Author mentions they use the Liar dataset, but does not mention which settings were considered and the source of info used. The paper fails to specify fundamental details such as the exact task formulation for each benchmark. For instance, whether the Liar dataset was used for binary or 6-class classification [3] , or what specific prompts and evaluation metrics were employed for summarization and program repair. Without a clear account of these implementation choices, the performance metrics and the resulting "chemistry" scores cannot be verified. This significantly impacts the paper soundness

6) Not clear how/why the 3 datasets used falls in the low, medium and high complexity category

7) Datasets used for testing (like Liar) were proposed before these LLMs were created and as such it is impossible to know whether they were part of their training data. This is a problem when doing research and assessing their performance on such standard benchmarks. Authors should comment on this problem [1] and consider techniques such as [2] to assess the contamination. To me the contamination becomes clear when the author mention a model saying exactly : “half-true, mostly-true, and true” (lines 48-52) which are exactly the labels used in the original liar dataset in which each statement is labeled as belonging to one of the following {pants-fire, false, barely- true, half-true, mostly-true, and true}.

8) Lines 34-36: (1) before inference, e.g., LLM routers (Rosenbaum et al., 2018); But the work by Rosenbaum was never based on LLMs. Just a routing. This is misleading.

9) Custom subset of models for each individual query is the defining characteristic of DES methods. The paper's failure to review this area and clearly differentiate its approach from existing DES techniques is a major scholarly oversight. Moreover, The goal of building a subset of models that complement each other's strengths and weaknesses is a long-standing objective in ensemble learning. The paper does not position its "chemistry" metric against these existing efforts.

10)	Continuing with the point above, some works like [5] also propose graph structure for modeling the he relationships between models and data instances and recommend the best ensembles which just adds to the point that this work need a more complete literature review on ensemble learning to proper potition their paper with the existing literature.

11)	Lines 495 to 500 -  the authors mention “By guiding efficient ensemble formation, our framework can help mitigate these costs and support sustainability.” However,if the "chemistry" for a specific query Q is calculated based on the cost_Q(X) of different model subsets, this logically requires generating outputs from multiple models (or all models) for that very query Q before the best ensemble can be selected. Thus, even the final ensemble is smaller than the full set, the cost (compute, energy) won’t be lower.

12)	Lastly, A motivation for multi-LLM collaboration is to surpass the capabilities of any single model. However, the evaluation exclusively compares different ensemble formation strategies (which, according to the author’s definitions, can even damage performance if not done properly) against each other. As such, they should consider a single model baseline as well (like the best single one for instance).

**Refs:**

[1] Dong, Yihong, et al. "Generalization or memorization: Data contamination and trustworthy evaluation for large language models." arXiv preprint arXiv:2402.15938 (2024).

[2] Yax, Nicolas, Pierre-Yves Oudeyer, and Stefano Palminteri. "Assessing contamination in large language models: introducing the LogProber method." arXiv preprint arXiv:2408.14352 (2024).

[3] Wang, William Yang. "" liar, liar pants on fire": A new benchmark dataset for fake news detection." arXiv preprint arXiv:1705.00648 (2017).

[4] Li, Yinheng. "A practical survey on zero-shot prompt design for in-context learning." arXiv preprint arXiv:2309.13205 (2023).

[5] Souza, Mariana A., et al. "A dynamic multiple classifier system using graph neural network for high dimensional overlapped data." Information Fusion 103 (2024): 102145.

**Questions:**

1)	How does “Chemistry” or the complimentary in this paper differ significantly from the notion of complemenraty in ensemble learning?

2)	How does the "Chemistry" metric differ significantly from established notions of model diversity and complementarity in the ensemble learning literature (considering both: static and dynamic selection)?

3)	Please specify the exact experimental setup for each benchmark (e.g., was Liar used for 2-class or 6-class classification? Which prompts were used for each task and models?. A detailed appendix on implementation details is crucial for soundness.

4)	Given that benchmarks like Liar predate the LLMs used, have you assessed potential data contamination? Please refer to the papers mentioned in the weakness section for measuring that.

5)	calculating cost_Q(X) for a query seems to require running multiple models on Q first. Can you clarify the operational workflow and how it avoids this inherent cost, making it truly more efficient than a fixed ensemble? Also can you provide the cost (FLOPS, inference time) of the proposal?

6)	How does the performance of your final "Chemistry"-recommended ensemble compare to a simple baseline of using the single best-performing LLM (e.g., GPT-4o) in isolation? Furthermore, what is the performance gap between your method and an "oracle" selector that always chooses the best possible model(s) for each query?

---

### Official Review · Reviewer_HaDC · 2025-10-28

**Soundness:** 2
**Presentation:** 2
**Contribution:** 2
**Rating:** 4
**Confidence:** 3

**Summary:**

The paper introduces LLM Chemistry, a framework that measures when LLM combinations exhibit synergistic or antagonistic behaviors that shape collective performance beyond individual capabilities. The analysis shows that chemistry among collaborating LLMs is most evident under heterogeneous model profiles, with its outcome impact shaped by task type, group size, and complexity.

**Strengths:**

- This work shifts the focus of multi-LLM collaboration from allocation to interaction. Rather than treating LLMs as isolated units, the paper argues that understanding and leveraging synergistic cooperation among multiple models is a more critical direction.
- The paper provides a clear and comprehensive definition of “LLM chemistry” and further incorporates optimization strategies for selecting complementary models.

**Weaknesses:**

- As shown in Table 1, the performance gains of the proposed method are quite limited. Notably, the approach even results in a ~30% performance degradation on one benchmark, which raises significant concerns about its practical effectiveness. The chemistry-based selection appears to improve performance only under specific conditions (e.g., certain tasks and ensemble sizes), while potentially harming performance when ensembles are small or already saturated.

- The rationale behind the quality score computation is unclear. The choice of MVLE as the scoring mechanism lacks justification.

- The experimental settings are somewhat narrow. All benchmarks used are relatively old, which could cause data contamination issues. Additionally, the experiments focus exclusively on very large models, leaving open questions regarding applicability to more resource-constrained settings.

- The framework does not consider computational cost, which is a critical factor in real-world deployment of multi-LLM systems. Ignoring such cost severely limits the practicality of the proposed method.

- Although the paper aims to shift the focus of multi-LLM collaboration from allocation to interaction, the actual design appears to treat all LLMs as identical units following the same role assumptions. This suggests the method still fundamentally relies on allocation rather than enabling richer interaction dynamics.

**Questions:**

n/a

---

### Official Review · Reviewer_zKce · 2025-10-31

**Soundness:** 1
**Presentation:** 2
**Contribution:** 2
**Rating:** 2
**Confidence:** 4

**Summary:**

The authors formalize the concept of "chemistry" between LLMs as the change in benefit when models are combined, proposing the CHEME algorithm to compute chemistry scores using Model Interaction Graphs (MIGs) and the RECOMMEND algorithm for optimal ensemble selection. Theoretical analysis demonstrates that chemistry emerges primarily under heterogeneous model performance profiles, with effects modulated by task type, complexity, and ensemble size. Empirical evaluation on the benchmarks provide evidence that chemistry-based selection can improve ensemble effectiveness in specific contexts.

**Strengths:**

1. The paper makes a notable contribution by formalizing LLM interactions through a principled cost function framework with proven properties. Theorem 1 and Corollary 1 provide counterintuitive insights that chemistry emerges only under heterogeneous performance profiles, establishing clear theoretical boundaries for when the approach is applicable.
2. The authors ground their novel concept in a solid mathematical framework, providing clear and formal definitions for cost, benefit, and chemistry.
3. Rather than overclaiming effectiveness, the authors honestly report mixed results and provide thoughtful analysis of when chemistry helps versus hurts (e.g., saturation effects in high-performing ensembles, task complexity moderation).

**Weaknesses:**

1. My biggest concern lies in the computational complexity of the proposed CHEME algorithm. It requires finding the maximum interaction effect over all possible subsets X in S\{a,b}. This complexity renders the method computationally intractable for any realistic scenario where the pool of candidate LLMs, $S$, is non-trivial (for example, |S| > 20)
2. Despite claiming "≈1,800 total experimental task executions," the actual dataset sizes are modest—only 10 records per trial across 10 trials means just 100 evaluation instances per configuration. For Liar benchmark (4,000 statements available), this represents 2.5% coverage. The high variance in results (e.g., Clinical Notes N=3 chemistry at 0.700 vs. baseline at 1.000) and the note about "temporary service issues" affecting claude-3-7 performance raise questions about result stability and whether observed differences reflect genuine chemistry effects or experimental noise.
3. The core cost formulation (Equation 1) uses arbitrary weights (1/i) and a 75%/25% weighting between generation and review accuracy that lack principled justification. The paper doesn't explore sensitivity to these hyperparameter choices, leaving uncertainty about whether results depend critically on these specific formulations.
4. The baseline selection strategies (Random, Performance, Local, Remote) are relatively simple. The paper doesn't compare against recent LLM routing methods (RouteLLM [1] , FrugalGPT [2] mentioned in related work), mixture-of-experts approaches, or learned selection (such as Smoothie [3] and RELM [4] which also uses a combination of model selection and evaluation) policies.

Refs:
1. Ong, Isaac, et al. "Routellm: Learning to route llms with preference data." arXiv preprint arXiv:2406.18665 (2024).
2. Chen, Lingjiao, Matei Zaharia, and James Zou. "Frugalgpt: How to use large language models while reducing cost and improving performance." arXiv preprint arXiv:2305.05176 (2023).
3. Guha, Neel, et al. "Smoothie: Label free language model routing." Advances in Neural Information Processing Systems 37 (2024): 127645-127672.
4. Kumar, Tarun, et al. "Co-Optimizing Recommendation and Evaluation for LLM Selection", ICLR 2025 Workshop on Foundation Models in the Wild.

**Questions:**

0. Authors are requested to respond to my comments in the weakness section above.
1. In line 119, the quality and accuracy scores are both defined over the same index $i$, but the quality is defined on the outputs, and accuracy is defined over the LLMs. Can you clarify this?
2. Why does equation 2 need to account for the benefit of a w.r.t. X? Shouldn't the benefit (or cost) of a be only dependent on b?
3. How much historical performance data is required to reliably construct MIGs and estimate chemistry? For a new task domain without existing performance histories, what cold-start strategy would you recommend?
4. How does the proposed RECOMMEND algorithm distinguish between beneficial (synergistic) and detrimental (antagonistic) high-chemistry pairs? Would a signed (non-absolute) metric that explicitly separates synergy from antagonism be more appropriate for a recommendation task?
5. The paper reports a negative correlation between chemistry and ensemble complementarity for medium- and high-complexity tasks, attributing this to "saturation effects." Could this finding also be interpreted as a failure of the chemistry metric itself, where it preferentially assigns high scores to redundant model pairs (whose individual benefits drop sharply when combined), thus actively selecting for lower complementarity?

---

### Author Response · Authors · 2025-12-03
**Thank you for your feedback!**

We thank all the reviewers for their insightful comments! We are working on addressing all of them and updating the paper accordingly. Unfortunately, due to reasons out of our control, we have not been able to interact with the reviewers up until now. Nonetheless, we appreciate your feedback and detailed comments!

---

### Meta-Review · Area_Chair_VqB3 · 2026-01-06

**Summary:**

This submission addresses a relevant direction in multi-LLM collaboration by introducing the "LLM Chemistry" framework to quantify synergistic/antagonistic model interactions. The work’s attempt to shift focus from individual model selection to collaborative dynamics is noteworthy, and the theoretical formalization of chemistry with supporting proofs constitutes a modest strength.
However, the paper faces fundamental and unresolved limitations that undermine its scientific rigor and practical relevance, as highlighted by multiple reviewers. Key concerns include: (1) Computational intractability of the proposed CHEME algorithm for realistic LLM pool sizes; (2) Insufficient experimental validation (small dataset coverage, unaddressed result instability, potential data contamination, and lack of transparency on prompts/implementation details); (3) Unjustified hyperparameter choices in core cost formulations; (4) Inadequate comparison with established ensemble learning metrics (e.g., prediction diversity) and state-of-the-art LLM routing methods, leaving the novelty of "LLM Chemistry" unsubstantiated; (5) Poor reproducibility due to missing code and data; (6) Disregard for critical real-world factors like computational cost, and narrow experimental scope (old benchmarks, limited model types).
The authors’ rebuttal merely acknowledges reviewer feedback without addressing these core issues. Given the breadth and severity of the concerns—particularly around methodological soundness, experimental rigor, and practical applicability—the submission does not meet ICLR’s standards for acceptance. We encourage the authors to address these limitations thoroughly in future work.

**Reviewer Concerns:**

All key concerns from reviewers remain unaddressed, including:
Computational intractability of the CHEME algorithm for realistic LLM pool sizes (Reviewer zKce).
Insufficient experimental validation (small dataset coverage, result instability, data contamination, vague setup details) (Reviewers zKce, HaDC, qnij, UtP3).
Unjustified hyperparameters in core formulations and lack of sensitivity analysis (Reviewer zKce).
Weak novelty (failure to distinguish "LLM Chemistry" from existing ensemble learning metrics/DES methods) (Reviewers zKce, qnij).
Poor reproducibility (missing code/data, unspecified prompts) (Reviewer qnij).
Disregard for real-world costs and narrow experimental scope (old benchmarks, limited model types) (Reviewers HaDC, UtP3).
Unrealistic theoretical assumptions (e.g., no cross-terms, pairwise focus) (Reviewer UtP3).
Lack of critical baselines (single best LLM, oracle selector) (Reviewer qnij).

**Reviewer Scores:**

NA

---

### Decision · Program_Chairs · 2026-01-26

Reject